# Impact of Dietary Crude Protein Level on Hepatic Lipid Metabolism in Weaned Female Piglets

**DOI:** 10.3390/ani11061829

**Published:** 2021-06-18

**Authors:** Ning Liu, Yun Ji, Ying Yang, Hai Jia, Xuemeng Si, Da Jiang, Yunchang Zhang, Zhaolai Dai, Zhenlong Wu

**Affiliations:** 1State Key Laboratory of Animal Nutrition, Department of Animal Nutrition and Feed Science, China Agricultural University, Beijing 100193, China; nuli982390@163.com (N.L.); jean500@163.com (Y.J.); cauvet2020@outlook.com (Y.Y.); jiahai@cau.edu.cn (H.J.); sisi_sxm@cau.edu.cn (X.S.); B20193040341@cau.edu.cn (D.J.); B20173040318@cau.edu.cn (Y.Z.); daizhaolai@cau.edu.cn (Z.D.); 2Beijing Advanced Innovation Center for Food Nutrition and Human Health, China Agricultural University, Beijing 100193, China

**Keywords:** piglet, crude protein, liver, lipid metabolism

## Abstract

**Simple Summary:**

It has been reported that a high crude protein diet could reverse the diet-induced lipid accumulation in the liver of mice and rodents. However, in vivo data supporting a functional role of a high crude protein diet on hepatic lipid metabolism-associated genes and proteins in weaned piglets is not available. In the present study, we aimed to provide a mechanistic insight into alterations in the hepatic lipid lipogenesis, lipolysis, oxidation, and gluconeogenesis in response to different dietary crude protein levels. Our results demonstrated that dietary crude protein could regulate hepatic lipid metabolism through regulating hepatic lipid lipogenesis, lipolysis, oxidation, and gluconeogenesis. The result indicated an important role of dietary crude protein in regulating hepatic lipid metabolism in weaned piglets.

**Abstract:**

Amino acids serve not only as building blocks for proteins, but also as substrates for the synthesis of low-molecular-weight substances involved in hepatic lipid metabolism. In the present study, eighteen weaned female piglets at 35 days of age were fed a corn- and soybean meal-based diet containing 20%, 17%, or 14% crude protein (CP), respectively. We found that 17% or 20% CP administration reduced the triglyceride and cholesterol concentrations, while enhanced high-density lipoprotein cholesterol (HDL-C) concentration in serum. Western blot analysis showed that piglets in the 20% CP group had higher protein abundance of hormone-sensitive triglyceride lipase (HSL) and peroxisome proliferator-activated receptor-γ coactivator 1α (PGC-1α), as compared with other groups. Moreover, the mRNA expression of sterol regulatory element binding transcription factor 1 (SREBPF1), fatty acid synthase (FASN), and stearoyl-CoA desaturase (SCD) were lower in the 17% or 20% CP group, compared with those of the piglets administered with 14% CP. Of note, the mRNA level of acetyl-CoA carboxylase alpha (ACACα) was lower in the 17% CP group, compared with other groups. Additionally, the mRNA level of lipoprotein lipase (LPL), peroxisome proliferator-activated receptor alpha α (PPARα), glucose-6-phosphatase catalytic subunit (G6PC), and phosphoenolpyruvate carboxykinase 1 (PKC1) in the liver of piglets in the 20% CP group were higher than those of the 14% CP group. Collectively, our results demonstrated that dietary CP could regulate hepatic lipid metabolism through altering hepatic lipid lipogenesis, lipolysis, oxidation, and gluconeogenesis.

## 1. Introduction

The liver, a pivotal organ in the metabolism, plays a central role in regulating lipid homeostasis [1,2]. The functional role of liver in lipid metabolism is to absorb free fatty acids from blood circulation, hereafter free fatty acids were re-esterified into triglycerides (TG), and then were secreted back into the blood circulation within triglyceride-rich very-low-density lipoprotein [3,4]. Therefore, intrahepatic lipid levels are predominantly affected by a balance of lipid metabolism, including lipogenesis, lipolysis, oxidation, secretion, and gluconeogenesis [5]. Hepatocytes, the main liver parenchymal cells, could synthesize triglyceride from fatty acids (FA) and store triglycerides primarily in the form of lipid droplets [6]. Excessive accumulation of TGs within the liver leads to the onset or progress of liver metabolic syndromes, such as nonalcoholic fatty liver disease (NAFLD), nonalcoholic steatohepatitis (NASH), and fibrosis [7], which are becoming a global health problem in adults and children [8]. An understanding of the underlying mechanisms responsible for TG accumulation in the liver would benefit the development of therapeutic strategies for patients affected by the liver diseases [6]. Of note, accumulating evidence has demonstrated that nutritional status especially dietary crude protein (CP) level could alter liver lipogenesis, lipolysis, oxidation, transportation, and gluconeogenesis, and contribute to physiological functions [9,10].

Animals fed a high CP level diet is associated with excretion of nitrogen, which is a major contributor to environmental pollution [11]. Therefore, low CP diets supplemented with crystalline amino acids have been extensively investigated as an effective strategy to reduce product cost, nitrogen excretion, and diarrhea in weaned piglets [12,13]. Over the past two decades, growing studies have prompted us to investigate the effects of low CP diet supplemented with or without amino acids on growth performance, intestinal metabolism, and gut microbiota [14,15,16]. Additionally, previous study has shown that high dietary CP significantly reduce adipocyte size, fat percentage, and backfat thickness in finishing pigs [11]. However, in vivo data supporting an impact of dietary CP level on hepatic lipid metabolism in weaned piglets, an animal model for studying human nutrition, are limited [17].

Amino acids serve not only as building blocks for proteins synthesis, but also as substrates for the synthesis of low-molecular-weight substances [18]. Indeed, emerging evidence showing that amino acids play a key role in preventing or ameliorating tissue TG accumulation in humans and mice [19,20]. For instance, L-arginine stimulates the expression of peroxisome proliferator-activated receptor-γ coactivator 1α (PGC-1α), nitric oxide synthase, heme-oxygenase, and AMP-activated protein kinase (AMPK). Moreover, L-arginine enhances lipolysis, the oxidation of glucose and fatty acids, and inhibits fatty acid synthesis at the whole-body level in rats [21,22]. Additionally, isoleucine has been reported to reduce TG concentrations in liver and muscle, and markedly enhance the mRNA expression of peroxisome proliferator-activated receptor alpha (PPARα) and uncoupling protein (UCP) in mice [20]. Also, relatively high CP diets supplemented with branched chain amino acids (BCAAs) have been reported to reduce the incidence of obesity and related metabolic disorders in rats [23,24]. In mammals, BCAAs are substrates for the synthesis of monomethyl branched chain fatty acids (mmBCFAs), which might implicate in and contribute to outcome of a lean liver [25].

Based on these findings, the objective of this study was to investigate the effect of CP level on the serum TG, cholesterol (CHO), and hepatic lipid metabolism-related gene expression in weaned piglets.

## 2. Materials and Methods

### 2.1. Piglets and Experimental Design

The experimental procedures were approved by Institutional Animal Care and Use Committee of China Agricultural University. A total of 18 crossbred healthy female piglets (Duroc × Landrace × Yorkshire) were weaned at 28 days of age. After 7 days of adaptation, piglets with the initial body of 9.57 ± 0.64 kg were randomly assigned to one of three groups (*n* = 6/group). Piglets were fed a corn- and soybean meal-based diet with different dietary CP levels (14%, 17%, and 20%). The piglets had free access to feed and water throughout the 45-day experimental period. The diets were formulated based on nutrient requirements of National Research Council (NRC; 2012) for the piglets. Additionally, lysine, methionine, tryptophan and threonine were supplemented to diets. Other amino acids and EAA/NEAA ratio of diets were detected and shown in Table 1.

### 2.2. Serum and Liver Collection

At the end of 45-day trial, jugular venous blood samples were obtained from each piglet. Serum was obtained in centrifugation 1500× *g* for 10 min after incubation at room temperature for 1 h. Serum was separated and stored at −80 °C in tubes for further analysis [26]. Then, piglets were killed by euthanasia as previous described [10]. Liver was quickly isolated and stored at −80 °C for the later analysis. From each liver, 18 tissue samples (each approximately 25 mm × 25 mm × 15 mm) were collected to represent left medial.

### 2.3. Triglyceride, Cholesterol, HDL-C, and LDL-C Determination

Concentrations of triglyceride and cholesterol in liver and serum were determined by using a commercial assay kits from Nanjing Jiancheng Biochemistry (Catalog no. A110-1-1 and A111-1-1; Nanjing, China). Concentrations of high-density lipoprotein cholesterol (HDL-C) and Low-density lipoprotein cholesterol (LDL-C) in serum were measured by using a commercial assay kits from Nanjing Jiancheng Biochemistry (Catalog no. A112-1-1 and A113-1-1; Nanjing, China).

### 2.4. Urea Determination

Concentrations of urea in serum were determined by using a commercial assay kit purchased from Nanjing Jiancheng Biochemistry (Catalog no. C013-2-1; Nanjing, China) based on previous describe [27].

### 2.5. Quantitative Real-Time PCR

Total RNA was isolated from the liver by using the Trizol reagent (CWBio Biotech Co., Beijing, China), and then the RNA was reverse transcribed into cDNA with a high-capacity cDNA archive kit (Applied Biosystems, CA, USA) according to the manufacturer’s protocol. The gene expression was examined by quantitative real-time PCR as previously described [28,29]. Primer sequences used for genes were designed using Primer 3 web software. The primer sequences (5′–3′) of sterol regulatory element binding transcription factor 1 (SREBPF1), fatty acid synthase (FASN), acetyl-CoA carboxylase alpha (ACACα), stearoyl-CoA desaturase (SCD), lipoprotein lipase (LPL), peroxisome proliferator-activated receptor gamma coactivator 1-alpha (PPARα), glucose-6-phosphatase catalytic subunit (G6PC), phosphoenolpyruvate carboxykinase 1 (PKC1), and glyceraldehyde-3-phosphate dehydrogenase (GAPDH) were listed in Table 2. The primer sequences (5′–3′) of 3-hydroxy-3-methylglutaryl coenzyme A reductase (HMGCR), microsomal triglyceride transfer protein (MTTP), apolipoprotein B (ApoB), carnitine palmitoyltransferase 1B (CPT1B), carbohydrate response element binding protein (ChREBP), and Leptin were listed in Appendix A. Fold change of mRNA level was calculated using the standard 2^−ΔΔCt^ method, and GAPDH was used as a reference gene for normalization [30].

### 2.6. Western Blot Analysis

Frozen liver was homogenized in a ceramic mortar with liquid N_2_ and then lysed in the ice-cold radioimmunoprecipitation assay lysis buffer containing 50 mmol/L Tris-HCl (pH 7.4), 150 mmol/L NaCl, 1% NP-40, 0.1% SDS, 1.0 mmol/L phenylmethanesulfonyl fluoride (PMSF), 1.0 mmol/L Na_3_VO_4_, 1.0 mmol/L NaF, and protease inhibitor cocktail (Roche, Indianapolis, IN, USA). Equal amounts of proteins (50 μg) were separated on 10% SDS-PAGE gels after determination of protein concentration with a bicinchoninic acid protein assay kit (Huaxingbio, Beijing, China), and then proteins were transferred to PVDF membranes (PVDF, Millipore, Billerica, MA, USA). The membranes were blocked in 5% fat-free milk solution for 1 h at 25 °C and then were incubated with indicated primary antibodies overnight at 4 °C. Antibodies against GAPDH (Catalog no. sc-59540), HSL (Catalog no. sc-25843), and PGC-1α (Catalog no. sc-13067) were obtained from Santa Cruz Biotechnology (Santa Cruz, CA, USA). After that, the members were incubated with a horseradish peroxidase-conjugated secondary antibody for 1 h at room temperature. The protein bands were incubated with an enhanced chemiluminescence kit (1:5000; Huaxingbio, Beijing, China) and developed by using the ImageQuant LAS 4000 mini system (GE Healthcare, Piscataway, NJ, USA). Density of protein bands was quantified by using the Image-Pro Plus 6.0 software (Media Cybernetics, CA, USA).

### 2.7. Statistical Analysis

All data were presented as means ± SEMs and were subjected to a one-way analysis of variance (ANOVA; SAS Version 9.1). Differences between group were determined by using the Tukey’s multiple comparisons test. Differences at *p* < 0.05 were considered significant.

## 3. Results

### 3.1. Effects of Dietary Crude Protein on Triglyceride and Cholesterol Concentration in Liver and Serum

As illustrated, 17% or 20% CP administration reduced the concentrations of TG (Figure 1A,B) and CHO (Figure 1C,D) in liver and serum when compared with those of piglet administered with 14% CP (*p* < 0.05). Concentrations of CHO in liver and serum were lower in 20% CP, as compared with 14% CP groups (Figure 1C,D). Additionally, no difference was observed in serum and liver TG concentration between 20% CP and 17% CP group (*p* > 0.05). Similarly, no difference was observed in liver CHO concentration between 14% CP group and 17% CP group (*p* > 0.05). Notably, piglets in the 17% or 20% CP group had a higher concentration of HDL-C (Figure 1E) in serum, when compared with 14% CP group (*p* < 0.05). Moreover, LDL-C concentration in serum was not affected by CP levels (*p* > 0.05).

### 3.2. Effects of Dietary Crude Protein on Hepatic Lipogenesis Genes Expression

mRNA expression of SREBPF1 and the downstream of its targets, such as FASN, ACACα, and SCD were determined. As shown in Figure 2, 17% or 20% dietary CP downregulated the mRNA expression of SREBPF1 (Figure 2A), FASN (Figure 2B) and SCD (Figure 2C) when compared with those of piglet administered with 14% CP (*p* < 0.05). Consistently, 17% CP dramatically reduced the mRNA expression of ACACα (Figure 2D) when compared with 14% CP group (*p* < 0.05). However, no difference was observed between 14% CP and 20% CP group (*p* > 0.05). Taken together, our results showed that dietary CP regulated the master genes expression which involved in hepatic lipogenesis and downstream targets genes.

### 3.3. Effects of Dietary Crude Protein on Hepatic Lipolysis and Lipid Oxidation

To investigate the protein abundance of HSL (Figure 3A) in response to different CP levels, western blot analysis was performed to measure liver protein abundance of HSL in weaned piglets. Protein abundance of HSL (Figure 3A) in liver was enhanced (*p* < 0.05) in piglets fed with 17% or 20% CP diet, compared with those of piglets fed with 14% CP diet. Consistently, mRNA expression of LPL (Figure 3C) was significantly upregulated by 20% CP when compared with 14% or 17% CP groups (*p* < 0.05). Additionally, protein abundance of PGC-1α (Figure 3B), mRNA expression of PPARα (Figure 3D) were significantly enhanced in 17% or 20% CP group when compared with 14% CP group (*p* < 0.05). Additionally, relative high dietary CP enhanced the mRNA expression of HMGCR (Appendix A), MTTP (Appendix A), ApoB (Appendix A), CPT1B (Appendix A), and ChREBP (Appendix A) when compared with 14% CP group. However, no difference was observed in the mRNA expression of Leptin (Appendix A). These results indicated that high dietary CP reduced liver TG concentration through the enhancement of lipolysis and lipid oxidation.

### 3.4. Effects of Dietary Crude Protein on Serum Urea Concentrations and Hepatic Gluconeogenesis

The concentrations of urea in serum (Figure 4A), as a major metabolic endpoint product of amino acids, were reduced (*p* < 0.05) in the 17% CP group when compared with the 14% CP group. However, 20% dietary CP enhanced (*p* < 0.05) the urea concentration in serum when compared with those of piglets fed with 14% or 17% CP diet (Figure 4A). To further understand the mechanism of dietary CP on liver lipid metabolism, mRNA expression of G6PC and PKC1 involved in gluconeogenesis were determined. mRNA expression of G6PC (Figure 4B) and PKC1 (Figure 4C) were upregulated in 20% CP group when compared with the 14% or 17% CP group (*p* < 0.05). However, no differences were observed in the mRNA expression of G6PC and PKC1 between the 14% and 17% CP groups (*p* > 0.05).

## 4. Discussion

Base on previous study, it has been well demonstrated that low dietary CP could reduce product costs, nitrogen excretion and diarrhea [31]. Mounting evidence indicates that low dietary CP reduced the growth performance by affecting intestinal morphology, digestive enzymes and gut microbiota in weaned piglets [11,13]. Furthermore, accumulating evidence suggest that high dietary CP reduced mRNA expression of glycolysis enzymes (GK, L-PK) and lipogenesis enzymes (ACACα, FASN), and upregulated mRNA expression of gluconeogenesis enzymes in rats [11,32]. To our knowledge, the effects and the underlying mechanism of dietary CP level on lipid metabolism in the liver of weaned piglets, the best model for studying human nutrition, have not been elucidated [1,2,33]. In the present study, we found that high dietary CP reduced TG concentration in liver and serum through regulating hepatic lipogenesis, lipolysis, oxidation, and gluconeogenesis (single data points are provided in the Appendix B).

Enhanced TG concentration in liver and blood circulation, reduced β-oxidation in the liver, and/or reduced synthesis or secretion of apolipoproteins are major determining factors in the development of liver metabolic syndrome [1]. Furthermore, accumulation of lipid infiltration leads to hepatic steatosis [34]. In the present study, TG and CHO concentrations (Figure 1A,B) in liver and serum were reduced in the 17% CP groups compared with piglets fed with 14% dietary CP. However, no difference was observed between the 17% and 20% CP group in TG concentration. Correspondingly, the reduced TG concentration was associated with downregulated mRNA level of SREBPF1, FASN, ACACα, and SCD (Figure 2). Similarly, previous studies have shown that high protein intake reduces hepatic lipid accumulation and plasma TG concentration in rats and humans [35,36], alleviates steatosis [37], and reduces body weight [38]. SREBPs is a master regulator of lipo- and sterol- genic gene transcription on lipid homeostasis of vertebrate cells [8,39]. ACACα plays a key role in the provision of the malonyl-CoA substrate for fatty acids biosynthesis. Downregulation of SREBPF1, FASN, and ACACα at mRNA level as observed in the present study is in agreement with previous studies showing that high CP diet reduced the expression of the lipogenic genes (ACACα, FASN, and SREBPF1) in rats or diet-induced obese rats [33,36]. SCD is a limiting enzyme in the synthesis of monounsaturated fatty acids required for normal rates of synthesis of TG, cholesterol esters, and phospholipids [40]. Recent studies indicated that SCD1^-/-^ mice have lower concentration of hepatic TG and cholesterol esters, and are resistant to the liver steatosis [40,41]. Consistently, mRNA expression of SCD in the present study was dramatically downregulated with the rise of dietary CP which is consistent with the previous study [36].

Accumulation of TG in the liver is dependent not only on lipogenesis, but also on lipolysis, oxidation, transportation, and gluconeogenesis [42]. We next determined the proteins and genes expression implicated in lipid metabolism, including HSL, PGC-1α, PPARα, LPL, G6PC, and PKC1. Interestingly, our results revealed that 20% dietary CP level markedly enhanced HSL protein abundance and mRNA expression of LPL (Figure 3A,C) when compared with 14% or 17% CP groups, suggesting that the enhancement of lipolysis by relative high dietary CP [43,44]. This finding was in agreement with the previous study [32]. The protein abundance of PGC-1α and mRNA expression of PPARα were also determined in the liver. As expected, our data indicated that relative high CP diet was associated with enhanced lipid oxidation [36].

G6PC and PKC1 are important molecules involved in gluconeogenesis and glycogenolysis. In the present study, we demonstrated that 20% CP enhanced G6PC and PKC1 mRNA expression when compared with the 14% or 17% CP groups [45]. In line with an enhanced mRNA expression of G6PC and PKC1, a pronounced enhancement in the circulating concentration of urea in serum were observed in piglets fed with 20% CP diet. Urea concentration in serum represents the utilization efficiency of protein, also conversion of amino acids into glucose [46,47,48]. In agreement with our findings, previous studies show that PCK1 expression was upregulated ~3-fold by a high CP diet [10,32]. However, the result of G6PC expression in our study is inconsistent with previous study [10]. The discrepancy between ours and previous study might be explained by using different animal models. Amino acids serve not only as building blocks for proteins, but also as substrates for the synthesis of low-molecular-weight substances which contribute to the hepatic lipid metabolism [18]. Furthermore, growing studies have demonstrated that some amino acids play a key role in preventing or ameliorating tissue TG accumulation [20,49]. L-arginine, BCAAs, lysine, threonine, and low-molecular-weight peptides have been reported to reduce liver TG concentration by modulating lipid metabolism and lipid metabolism-related gene expression [50,51].

## 5. Conclusions

Collectively, our results provided important evidence for a dietary CP level in regulating hepatic lipid metabolism in weaned piglets. A dietary CP level could regulate hepatic lipid metabolism through regulating hepatic lipid lipogenesis, lipolysis, oxidation, and gluconeogenesis.

## Figures and Tables

**Figure 1 animals-11-01829-f001:**
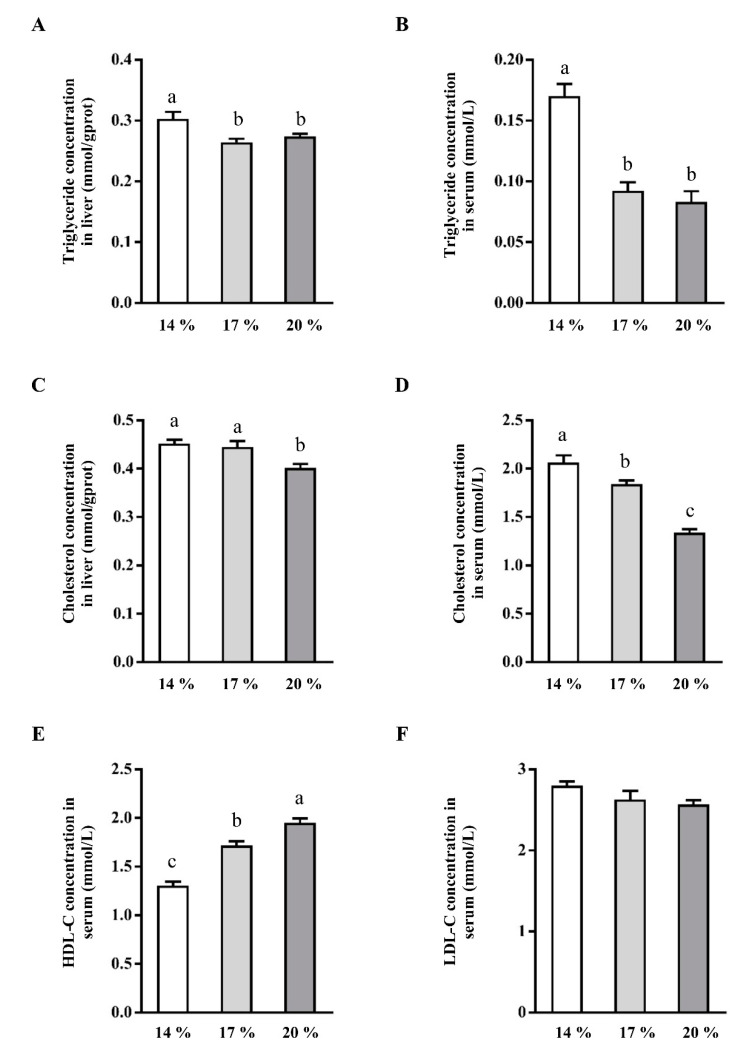
Effect of dietary crude protein level on the concentrations of triglyceride and cholesterol in liver and serum. Triglyceride concentration in liver (**A**) and serum (**B**). Cholesterol concentration in liver (**C**) and serum (**D**). HDL-C (**E**) and LDL-C (**F**) concentration in serum. Values are means ± SEMs, *n* = 6. Means without a common letter differ, *p* < 0.05. HDL-C, high-density lipoprotein cholesterol; LDL-C, low-density lipoprotein cholesterol.

**Figure 2 animals-11-01829-f002:**
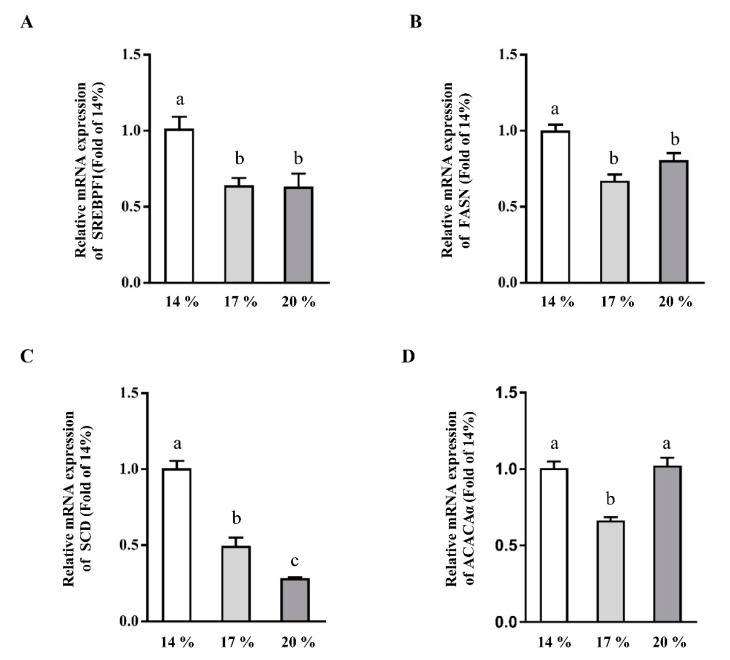
Effects of dietary crude protein level on hepatic lipogenesis genes expression. SREBPF1 (**A**), FASN (**B**), SCD (**C**), and ACACα (**D**) mRNA expression in liver of weaned piglets fed different dietary crude protein. Values are means ± SEMs, *n* = 6. Means without a common letter differ, *p* < 0.05. SREBPF1, sterol regulatory element binding transcription factor 1; FASN, fatty acid synthase; ACACα, acetyl-CoA carboxylase alpha; SCD, stearoyl-CoA desaturase.

**Figure 3 animals-11-01829-f003:**
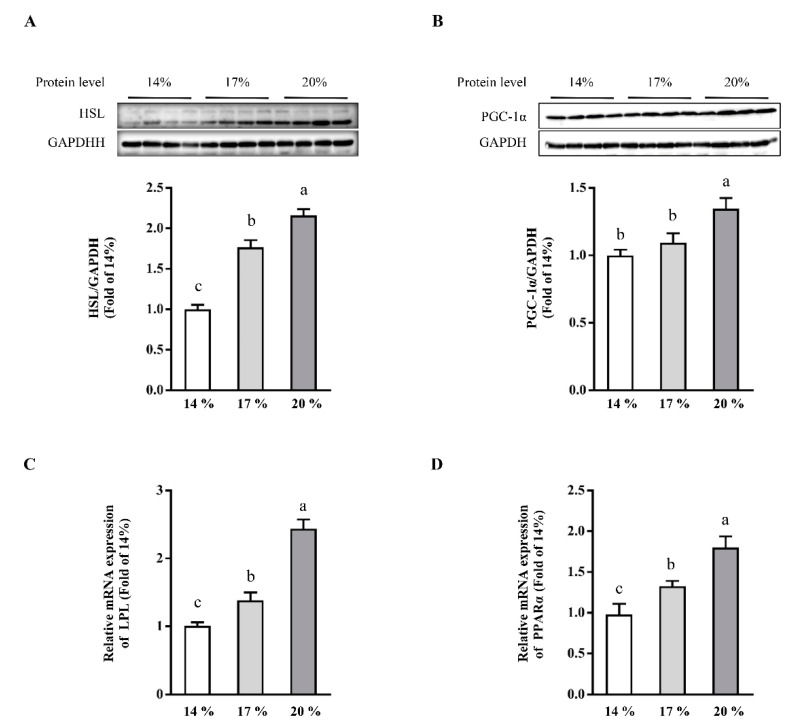
Effects of dietary crude protein level on hepatic lipolysis and lipid oxidation. Immunoblot analysis of HSL (**A**) and PGC-1α (**B**) protein abundances, LPL (**C**), and PPARα (**D**) mRNA abundance in the liver of weaned piglets fed different dietary crude protein. Values are means ± SEMs, *n* = 4 (protein abundance) or 6 (mRNA expression). Means without a common letter differ, *p* < 0.05. HSL, hormone-sensitive lipase; PGC-1α, peroxisome proliferator-activated receptor gamma coactivator 1-alpha; LPL, lipoprotein lipase; PPARα, peroxisome proliferator-activated receptor alpha.

**Figure 4 animals-11-01829-f004:**
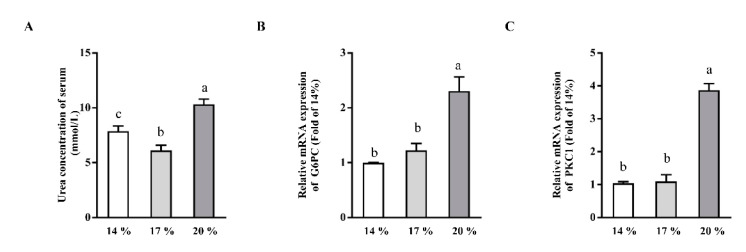
Effects of dietary crude protein level on serum urea concentration and hepatic gluconeogenesis. Urea (**A**) concentration in serum, and G6PC (**B**) and PKC1 (**C**) mRNA abundance in the liver of weaned piglets fed different dietary crude protein. Values are means ± SEMs, *n* = 6. Means without a common letter differ, *p* < 0.05. G6PC, glucose-6-phosphatase catalytic subunit; PKC1, phosphoenolpyruvate carboxykinase 1.

**Table 1 animals-11-01829-t001:** Crude protein content and amino acids composition of the experimental diets ^1^.

Item	Content, %		
	14% CP	17% CP	20% CP
Crude protein	14.14	17.32	20.17
EAA ^2^			
Arginine	0.71	0.93	1.09
Histidine	0.30	0.37	0.44
Isoleucine	0.46	0.60	0.71
Leucine	1.11	1.32	1.52
Lysine	1.26	1.25	1.26
Methionine	0.41	0.42	0.40
Methionine + Cystine	0.63	0.65	0.62
Phenylalanine	0.56	0.70	0.81
Threonine	0.76	0.75	0.76
Tryptophan	0.20	0.20	0.20
Tyrosine	0.41	0.50	0.59
Valine	0.54	0.64	0.72
NEAA ^3^			
Alanine	0.75	0.90	1.07
Asparagine	1.15	1.49	1.76
Cystine	0.22	0.23	0.22
Glutamate	2.28	2.78	3.15
Glycine	0.53	0.71	0.92
Proline	0.90	1.04	1.17
Serine	0.60	0.74	0.85
EAA	6.29	7.18	7.91
NEAA	6.84	8.40	9.74
EAA/NEAA	0.92	0.85	0.81
Calculated nutritional value			
DE ^4^ (MJ/kg)	14.60	14.60	14.60
Total calcium	0.70	0.71	0.69
Total phosphorus	0.53	0.55	0.57

^1^ Values are the means of a chemical analysis conducted in duplicate. ^2^ essential amino acid. ^3^ nonessential amino acid. ^4^ digestible energy.

**Table 2 animals-11-01829-t002:** Primers used for real-time PCR.

Genes	Accession No.	Primers
*SREBPF1*	NM_214157.1	F: 5′-GACCCCACCAGTCCTGATG-3′
		R: 5′-ACGGGTACATCTTCAGCGG-3′
*FASN*	NM_001099930.1	F: 5′-GTTCCAAGGAGCAAGGTGTG-3′
		R: 5′-GCTTCGATGTACTCCAGGGA-3′
*ACACα*	NM_001114269.1	F: 5′-ATGTCTGGCTTGCACCTAGT-3′
		R: 5′-ATAAGACCACCGGCGGATAG-3′
*SCD*	NM_213781.1	F: 5′-AGAAGACATCCGCCCTGAAA-3′
		R: 5′-TCTTGCAGGTGGGGATCAAT-3′
*LPL*	NM_214286.1	F: 5′-CGCGGACAGAATTTCAGGAG-3′
		R: 5′-GGCAAGTGTCCTCAACTGTG-3′
*PPARα*	NM_001044526.1	F: 5′-GCAAGCTTGGACTTGAACGA-3′
		R: 5′-GCATCCCGTCCTTGTTCATC-3′
*G6PC*	NM_001113445.1	F: 5′-TGTGGGCATCAAACTCCTCT-3′
		R: 5′-GCTTTATCAGTGGCACCGAG-3′
*PKC1*	FJ668384.1	F: 5′-CGTTTACTGGGAAGGCATCG-3′
		R: 5′-TTCCCCTACAACAGCCAGAG-3′
*GAPDH*	NM_001206359.1	F: 5′-GTCGGAGTGAACGGATTTGG-3′
		R: 5′-AGTGGAGGTCAATGAAGGGG-3′

## Data Availability

Data are contained within the article or Appendix A.

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
