# Peer review of "Impact of Dietary Crude Protein Level on Hepatic Lipid Metabolism in Weaned Female Piglets"

_animals, 2021, doi:10.3390/ani11061829_

Round 1

Reviewer 1 Report

The subject of the manuscript is interesting. The introduction sufficiently describes the basics of the research. The research hypothesis has been formulated well, but there is no clearly defined research goal.

In my opinion, it is necessary to supplement the publication with a table with a detailed composition of the feed.

The results should be supplemented with information on the amount of feed consumed. Metabolic differences may result not only from differences in feed composition but also from differences in consumption, if any.

The sentence in lines 171-172 and 176-177 sounds like a discussion and should be moved there.

Author Response

Reviewer#1

Comments and Suggestions for Authors

The subject of the manuscript is interesting. The introduction sufficiently describes the basics of the research. The research hypothesis has been formulated well, but there is no clearly defined research goal.

Reply:

Thanks for reviewing our manuscript and the constructive comments. We clearly defined our research goal in this version. Your comments and suggestions are of great value for our study.

  1. In my opinion, it is necessary to supplement the publication with a table with a detailed composition of the feed.

Reply:

Thanks for your suggestion, the composition of diet was present in published paper based on the shared projects (Wu et al. / J Zhejiang Univ-Sci B (Biomed & Biotechnol) 2015 16(6):496-502).

  1. The results should be supplemented with information on the amount of feed consumed. Metabolic differences may result not only from differences in feed composition but also from differences in consumption, if any.

Reply:

The growth performance of piglets was present in published paper based on the shared projects (Wu et al. / J Zhejiang Univ-Sci B (Biomed & Biotechnol) 2015 16(6):496-502).

  1. The sentence in lines 171-172 and 176-177 sounds like a discussion and should be moved there.

Reply:

We moved those sentences as you suggested. Thanks again!

Reviewer 2 Report

Review animals-1191254. "Dietary protein level alters hepatic lipid metabolism in weaned piglets" concerns the issue of liver metabolism in response to differences in a diet containing variable CP content. The manuscript is interesting because it presents changes in gene expression of the most important genes involved in lipid lipogenesis, lipolysis, oxidation, and gluconeogenesis and also concentrating Triglyceride, cholesterol, HDL-C, and LDL-C in the serum and liver and urea in the serum. The manuscript is well written in my opinion. However, the English language is not my native. Nevertheless in this manuscript is missing information about the health of piglets dependent on CP content in the diet I think that is important point. Maybe these diets were tested before and the poofs can be cited that these diets do not harm or may the authors have phenotypic data confirm this assumption to consider.  

Below minor coments:

Material and methods:

 In table 2 primers for GAPDH were missing. Why this endogenous control was selected?

Because the authors used non-specific qPCR signal detection using syber green as a fluorescent label. Was it applied for each analysed gene any negative control  (no-RT with or reaction mix without cDNA samples), because it is common that in late cycles in this kind of reaction primers form duplicates and fluorescent signal increases despite the lack of target cDNA.Un fortunatly the Figures are barely visible.

As above. In this study in my opinion is a lack of element concerning the health of these piglets maintained on lower and higher protein diets. Because in human, high protein diet with low- fat often leads to the problem with removing amino acids from the body and with disorders in the kidney function and is associated with acidification of the organism. The authors should add some information about the health of these piglets. Maybe other publications concerned about this issue before and it is known that diet containing 20% CP does not cause any harm on piglet organism. Authors can also compare some phenotypic trait data between piglets on different diets. However in this kind of experiment the urine composition should be also analysed.

Author Response

Reviewer#2

Comments and Suggestions for Authors

Review animals-1191254. "Dietary protein level alters hepatic lipid metabolism in weaned piglets" concerns the issue of liver metabolism in response to differences in a diet containing variable CP content. The manuscript is interesting because it presents changes in gene expression of the most important genes involved in lipid lipogenesis, lipolysis, oxidation, and gluconeogenesis and also concentrating Triglyceride, cholesterol, HDL-C, and LDL-C in the serum and liver and urea in the serum. The manuscript is well written in my opinion. However, the English language is not my native. Nevertheless in this manuscript is missing information about the health of piglets dependent on CP content in the diet I think that is important point. Maybe these diets were tested before and the poofs can be cited that these diets do not harm or may the authors have phenotypic data confirm this assumption to consider.

Reply:

Thanks for reviewing our manuscript and the constructive comments. Your comments and suggestions are of great value for our study. Previous study demonstrated that high dietary crude protein (20.9%) does not harm the health of piglets(Zhang et al. / Amino Acids. 2013 Nov;45(5):1191-205. doi: 10.1007/s00726-013-1577-y.).

  1. In table 2 primers for GAPDH were missing. Why this endogenous control was selected?

Because the authors used non-specific qPCR signal detection using syber green as a fluorescent label. Was it applied for each analysed gene any negative control (no-RT with or reaction mix without cDNA samples), because it is common that in late cycles in this kind of reaction primers form duplicates and fluorescent signal increases despite the lack of target cDNA.

Unfortunatly the Figures are barely visible.

Reply:

Thank you for your suggestions. GAPDH is a housekeeping gene whose expression level do not differ among samples. Comparison of the CT value of a target gene with that of the endogenous control gene allows the gene expression level of the target gene to be normalized to the amount of input RNA or cDNA. This is done without determining the exact amount of template used in the reaction. The use of an endogenous control gene corrects for variation in RNA content, variation in reverse-transcription efficiency, possible RNA degradation or presence of inhibitors in the RNA sample, variation in nucleic acid recovery, and differences in sample handling. Additionally, Figures have been replaced with higher resolution images as suggested, please see Figures in this version for details.

  1. As above. In this study in my opinion is a lack of element concerning the health of these piglets maintained on lower and higher protein diets. Because in human, high protein diet with low- fat often leads to the problem with removing amino acids from the body and with disorders in the kidney function and is associated with acidification of the organism. The authors should add some information about the health of these piglets. Maybe other publications concerned about this issue before and it is known that diet containing 20% CP does not cause any harm on piglet organism. Authors can also compare some phenotypic trait data between piglets on different diets. However in this kind of experiment the urine composition should be also analysed.

Reply:

The piglets fed diets with 14%, 17% and 20% crude protein level throughout the 45 days experimental period appeared normal and healthy. Previous study demonstrated that high dietary crude protein (20.9%) does not harm the health of piglets(Zhang et al. / Amino Acids. 2013 Nov;45(5):1191-205. doi: 10.1007/s00726-013-1577-y.). We are sorry for not being able to provide the urine composition data in our study. We tried to collect urine to test the urine composition, which are critical for our study. Unfortunately, the urine collection was failed, so we did not provide the result in this version.

Reviewer 3 Report

The manuscript describes the results of a feeding trial of weaned piglets. Different levels of crude protein were tested for piglet liver traits. Authors reported on some serum parameters and selected a few candidate genes involved in mitochondrial oxidation and lipid metabolism. Unfortunately, performance parameters (feed intake, live weight, …) are missing. For nutritional aspects, the study objectives are clear; however, there is a considerable number of major weaknesses. Despite the lack of mechanistic insight, the pig is certainly not an appropriate model for infant liver health in terms of fatty liver! Unlike in human and rodents, liver has only a minor role in systemic lipid metabolism in the pig. Further, the feed composition is missing – dietary protein might have been replaced by e.g. starch, which strongly affect the nutrient flows in the pig including liver function. Thus, whenever authors state the high protein content they need also to mention the low carbohydrate content.

General points to be considered:

  • Title: State the sex of pigs. “.. in weaned female piglets”). Further, I suggest to tune down the statement as the experimental output is rather weak (“impact on” instead of “alter”). Also include the protein-starch ratio in the title (“Dietary protein-starch ratio” instead of “Dietary protein level”)!
  • Simple summary line 15 and elsewhere in the manuscript: The authors stated that piglets serve as a model for infants. This might be the case in terms of other features but certainly not for systemic lipid metabolism. This has to be removed/rephrased.
  • Simple Summary line 19: This refers to “novel” findings which should be deleted since there is a large body of evidence that nutrient flows affect liver function.
  • Abstract line 24: include sex of piglets. Otherwise authors generalize effects, which is not appropriate.
  • Abstract line 32: It is known from several papers that urea levels increase in animals fed a high protein diet. The manuscript lacks interpretation why urea values are reduced in their trial.
  • Introduction lines 56-63: this very short paragraph is referring to literature on dietary protein content in the pig. The literature has to be searched more carefully to account for existing work.
  • Introduction lines 64-79: Add the species for each statement/cited manuscript.
  • Material & Methods: Authors should provide performance data, including live weight and feed intake as a minimum.
  • Material & Methods: Are the 18 piglets related?
  • Material & Methods Table 1: Details on feed composition is a must in these type of studies. Further, analytics have to be stated for at least crude ash, crude fat, starch, crude fibre.
  • Material & Methods line 102: Did authors stored blood at 37°C? I never heard of this – should be stored at room temperature.
  • Material & Methods line 104: Which part of the liver was sampled?
  • Material & Methods lines 107-116: State catalog numbers of the used kits to increase plausibility. State reference values / range of the kits when not designed for an animal lab. What are reference values for the pig in other cohorts?
  • Material & Methods Table 2: Missing primers for GAPDH.
  • Material & Methods lines 148-151: Why the Duncan approach has been used instead of Tukey post-hoc test statistics? The Duncan approach is known to produce false-positive results. Further, authors did not correct for litter – please explain.
  • Results lines 156-158: Should be rephrased and needs more explanation, “similar” is somehow misleading.
  • Results Figure 1-3: How SEM were calculated? Figure resolution is very poor, hard to read at this stage. Please provide graphs with single data points as manuscript appendix.
  • Results Figure 1: C and D should be re-ordered.
  • Results Figure 2: No idea how authors calculated that expression values. Please refer to this in the M&M section. I would expect a heatmap stating copy numbers. It is strongly encouraged to provide single data points in the appendix of the manuscript.
  • Discussion lines 264-266: refers to which group? Controls?
  • Conclusion: Remove “novel”. Might be possible to say, that results are somehow complementing existing literature in other species.
  • Conclusion line 291: Statement is misleading and interprets by far too much in the results! Since no assessment of phenotype is provided (histology) and simply no severe phenotype has been induced beforehand, this statement has to be deleted.

Additional points:

  • Language needs to be checked.
  • References: Note the layout for references 1-8.

Author Response

Reviewer#3

Comments and Suggestions for Authors

The manuscript describes the results of a feeding trial of weaned piglets. Different levels of crude protein were tested for piglet liver traits. Authors reported on some serum parameters and selected a few candidate genes involved in mitochondrial oxidation and lipid metabolism. Unfortunately, performance parameters (feed intake, live weight, …) are missing. For nutritional aspects, the study objectives are clear; however, there is a considerable number of major weaknesses. Despite the lack of mechanistic insight, the pig is certainly not an appropriate model for infant liver health in terms of fatty liver! Unlike in human and rodents, liver has only a minor role in systemic lipid metabolism in the pig. Further, the feed composition is missing – dietary protein might have been replaced by e.g. starch, which strongly affect the nutrient flows in the pig including liver function. Thus, whenever authors state the high protein content they need also to mention the low carbohydrate content.

Reply:

Thanks for reviewing our manuscript and the constructive comments. Your comments and suggestions are of great value for our study. The piglets fed diets with 14%, 17% and 20% crude protein level throughout the 45 days experimental period appeared normal and healthy. Our point-by-point responses are shown below and heighted in revised version.

General points to be considered:

  1. Title: State the sex of pigs. “.. in weaned female piglets”). Further, I suggest to tune down the statement as the experimental output is rather weak (“impact on” instead of “alter”). Also include the protein-starch ratio in the title (“Dietary protein-starch ratio” instead of “Dietary protein level”)!

Reply:

Thank you for your suggestions. According to for your suggestions, the title has

been modified in this version.

  1. Simple summary line 15 and elsewhere in the manuscript: The authors stated that piglets serve as a model for infants. This might be the case in terms of other features but certainly not for systemic lipid metabolism. This has to be removed/rephrased.

Reply:

Scientific evidence supports this similarity that most systems and organs have between pigs and human beings. The pig has long been an animal of interest for liver, kidney, pancreas, or islet and heart transplantations (Lawrence B Schook et al. / Annu Rev Anim Biosci. 2015; 3:219-44.). Systemic lipid metabolism between human and pig are certainly different, however, lipid metabolism in liver is similarity. Indeed, we change infant to human in order to avoid the confusion.

  1. Simple Summary line 19: This refers to “novel” findings which should be deleted since there is a large body of evidence that nutrient flows affect liver function.

Reply:The suggested change was made.

  1. Abstract line 24: include sex of piglets. Otherwise authors generalize effects, which is not appropriate.

Reply:

We are sorry for missing information. We added the sex of piglets as suggested.

  1. Abstract line 32: It is known from several papers that urea levels increase in animals fed a high protein diet. The manuscript lacks interpretation why urea values are reduced in their trial.

Reply:

The concentrations of urea in the plasma which are the major metabolic endpoint products of amino acids. Blood urea nitrogen could be used to predict nitrogen excretion and efficiency of nitrogen utilization in pigs. Therefore, in current study, piglets were provided low-protein diets supplemented with four essential amino acids (Lys, Met, Thr and Trp) and blood urea nitrogen was used as an indicator of nitrogen utilization. 14% crude protein and 17% crude protein diet decrease urea concentration indicating higher efficiency of amino acids than 20% crude protein diet. This result is consistent with Lordelo et al. (2008) who revealed that nitrogen excretion from faeces and urine was reduced in piglets fed 17% CP diets compared with 20% CP diet supplemented with Lys, Met, Thr and Trp.

  1. Introduction lines 56-63: this very short paragraph is referring to literature on dietary protein content in the pig. The literature has to be searched more carefully to account for existing work.

Reply:

Sorry for the mistake. We combined line 56-63 and line 64-78 into one paragraph.

  1. Introduction lines 64-79: Add the species for each statement/cited manuscript.

Reply:

Thank you for your suggestions. We have added species for the statement.

  1. Material & Methods: Authors should provide performance data, including live weight and feed intake as a minimum.

Reply:

The growth performance of piglets was present in published paper based on the shared project (Wu et al. / J Zhejiang Univ-Sci B (Biomed & Biotechnol) 2015 16(6):496-502).

  1. Material & Methods: Are the 18 piglets related?

Reply:

Sorry for that. Piglets used in this study are not related.

  1. Material & Methods Table 1: Details on feed composition is a must in these type of studies. Further, analytics have to be stated for at least crude ash, crude fat, starch, crude fibre.

Reply:

The growth performance of piglets was present in published paper based on shared project (Wu et al. / J Zhejiang Univ-Sci B (Biomed & Biotechnol) 2015 16(6):496-502). Composition of diet was also present in this published paper.

  1. Material & Methods line 102: Did authors stored blood at 37°C? I never heard of this – should be stored at room temperature.

Reply:

Sorry for the mistake, the blood was stored at room temperature for a clot to form, please see details in this section.

  1. Material & Methods line 104: Which part of the liver was sampled?

Reply:

From each liver, 18 tissue samples (each approximately 25 × 25 × 15 mm) were collected to represent left media.

  1. Material & Methods lines 107-116: State catalog numbers of the used kits to increase plausibility. State reference values / range of the kits when not designed for an animal lab. What are reference values for the pig in other cohorts?

Reply:

Thanks for your suggestion, we added more information about the kits in the materials in this version.

  1. Material & Methods Table 2: Missing primers for GAPDH.

Reply:

We added primer sequence for GAPDH, please see details in Table 2.

  1. Material & Methods lines 148-151: Why the Duncan approach has been used instead of Tukey post-hoc test statistics? The Duncan approach is known to produce false-positive results. Further, authors did not correct for litter – please explain.

Reply:

We are sorry for our mistake. Differences between means were determined by using Tukey's multiple comparisons test. We have corrected it in the statistical analysis section. An individual weaned piglet was considered as the experimental unit in all statistical analyses. So, we did not correct for litter

  1. Results lines 156-158: Should be rephrased and needs more explanation, “similar” is somehow misleading.

Reply:

Sorry for the confusion,we rephrased sentence in this version to make it clear.

  1. Results Figure 1-3: How SEM were calculated? Figure resolution is very poor, hard to read at this stage. Please provide graphs with single data points as manuscript appendix.

Reply:

We are sorry for the unclear statement. SEM is calculated by taking the standard deviation and dividing it by the square root of the sample size. The Figures have been replaced with higher resolution images as suggested, please see Figures in this version for details. Furthermore, we provided graphs with single data points in manuscript appendix.

  1. Results Figure 1: C and D should be re-ordered.

Reply:

Sorry for the mistake, we re-ordered figure 1 in the revised version.

  1. Results Figure 2: No idea how authors calculated that expression values. Please refer to this in the M&M section. I would expect a heatmap stating copy numbers. It is strongly encouraged to provide single data points in the appendix of the manuscript.

Reply:

Fold change of mRNA level was calculated using the standard 2-ΔΔCt method, and GAPDH was used as a reference gene for normalization. Furthermore, we provided graphs with single data points as manuscript appendix.

  1. Discussion lines 264-266: refers to which group? Controls?

Reply:

Sorry for the confusion, we have modified the sentences in the discussion section to make it clear.

  1. Conclusion: Remove “novel”. Might be possible to say, that results are somehow complementing existing literature in other species.

Reply:

Thank you for your suggestions. We have corrected it as suggested.

  1. Conclusion line 291: Statement is misleading and interprets by far too much in the results! Since no assessment of phenotype is provided (histology) and simply no severe phenotype has been induced beforehand, this statement has to be deleted.

Reply:

We are sorry for the unclear statement.

  1. Language needs to be checked.

Reply:

Thanks for the suggestion on language. We have done the language polishing as request. Typos and grammar errors in the manuscript have been revised in this version.

  1. References: Note the layout for references 1-8.

Reply:

Sorry for the mistake, we have corrected the layout for reference 1-8.

Round 2

Reviewer 2 Report

Dear authors, 

please add the citation that GAPDH is a good endogenous control for liver expression or any Genorm results that confirm its expression is stable in this tissue. Besides, my suggestions were addressed, and it was indicated that the presented diets were healthy for piglets, and citation was added.

Author Response

Reviewer#2

  1. please add the citation that GAPDH is a good endogenous control for liver expression or any Genorm results that confirm its expression is stable in this tissue. Besides, my suggestions were addressed, and it was indicated that the presented diets were healthy for piglets, and citation was added.

Reply:

Thanks for reviewing our manuscript and the constructive comments. Previous demonstrated that GAPDH is a good endogenous control for liver expression (Selection of reference genes for gene expression studies in porcine hepatic tissue using quantitative real-time polymerase chain reaction). Additionally, diets used in the study were formulated based on nutrient requirements of National Research Council (NRC; 2012) and were healthy for piglets in previous studies. Thanks again!

Reviewer 3 Report

The following comments should still be considered:

  • About urea measurements: Still, what is your reasoning that urea levels are decreased in 17% compared to 14% protein level?
  • Line 262; line 266: Please interpret – it is known that there is no de-novo synthesis of fatty acids in the pig.
  • Figures are still of poor quality, this needs to be addressed.

Author Response

Reviewer#3

  1. About urea measurements: Still, what is your reasoning that urea levels are decreased in 17% compared to 14% protein level?

Reply:

Thanks for reviewing our manuscript and the constructive comments. Your comments and suggestions are of great value for our study. Beside dietary CP levels, blood urea concentration was also dependent upon a balance of amino acids. Our results indicated that low dietary crude protein level resulted in lower efficiency of nitrogen utilisation in pigs when compared with 17% crude protein level. Low protein level diet did not meet the growth and development of piglets.

  1. Line 262; line 266: Please interpret – it is known that there is no de-novosynthesis of fatty acids in the pig.

Reply:

For dogs, cats, pigs, cattle, sheep, and goats, de novo synthesis of fatty acids is centered in adipose tissues, whereas in humans and rodents, the liver is the primary site for de novo synthesis of fatty acids. In the present study (Figure 2), 17% or 20% dietary CP downregulated the mRNA expression of SCD (Figure 2C) when compared with these of piglet administered with 14% CP (p <0.05). Indicating a suppression of SCD expression in liver of pig.

  1. Figures are still of poor quality, this needs to be addressed.

Reply:

We do not know what happed to the images when were transferred into PDF. The Figures have been replaced with higher resolution images as suggested in last version, please see Figures for details in Word Version.